# Is There a Place for Brachytherapy in Vulvar Cancer? A Narrative Review

**DOI:** 10.3390/cancers15235581

**Published:** 2023-11-25

**Authors:** Sofia Cordoba, Abel Cordoba, Beatriz Gil, Raquel Benlloch, Silvia Rodriguez, Dina Najjari-Jamal, Sofía Santana, Lucie Bresson, Cristina de la Fuente, Jesús Romero

**Affiliations:** 1Department of Radiation Oncology, Puerta de Hierro University Hospital, 28222 Majadahonda, Madrid, Spain; bgilh@salud.madrid.org (B.G.); raquel.benlloch@salud.madrid.org (R.B.); sofia.santana@salud.madrid.org (S.S.); cristina.fuente@salud.madrid.org (C.d.l.F.); 2Department of Radiation Oncology, Oscar Lambret Comprehensive Cancer Center, 59000 Lille, France; a-cordoba@o-lambret.fr; 3Department of Radiation Oncology, Clinica Benidorm Hospital, 30501 Benidorm, Alicante, Spain; atosrodriguez@gmail.com; 4Department of Radiation Oncology, Catalan Institut of Oncology, University of Barcelona, Hospitalet de Llobregat, 08908 Barcelona, Spain; dnajjari@iconcologia.net; 5Department of Surgical Oncology, Henin-Beaumont Polyclinic, 62110 Henin Beaumont, France; lbresson@ahnac.com

**Keywords:** vulvar cancer, radiation therapy, brachytherapy, radiobiology

## Abstract

**Simple Summary:**

In view of the high recurrence rates of vulvar cancer, often even after successful surgery, this article emphasises the crucial role of radiotherapy. Surgery remains the primary treatment, but radiotherapy, particularly brachytherapy (BT) either alone or in combination with external beam radiotherapy (EBRT), plays an important role in the adjuvant and primary treatment of vulvar cancer. The article highlights the advantages of BT, including improved dose precision, minimised impact on organs at risk and support for hypofractionated accelerated treatment. This narrative review provides recent data on the importance of BT in the treatment of primary and recurrent vulvar cancer, covering radiobiological, clinical and therapeutic aspects.

**Abstract:**

Vulvar cancer is a relatively rare neoplasm. The essential treatment is surgery for the primary tumour. However, postoperative recurrence rates are high, even in early-stage disease when tumour-free surgical margins are achieved or in the absence of associated risk factors (lymph node metastases, deep stromal invasion or invasion of the lymphatic vascular space). Radiotherapy plays an important role in the treatment of vulvar cancer. Adjuvant treatment after surgery as well as primary treatment of locally advanced vulvar cancer (LAVC) is composed of two key radiotherapy treatment scenarios, external beam radiation therapy (EBRT) either combined or not combined with brachytherapy (BT). In a recurrence setting, where surgery is not an option, BT alone or in combination with EBRT can be used. Compared to EBRT, BT has the radiobiological potential to improve dose to the target volume, minimise the dose to organs at risk, and facilitate hypofractionated-accelerated treatment. This narrative review presents recent data on the role of BT in the treatment of primary and/or recurrent vulvar cancer, including radiobiological, clinical, and therapeutic aspects.

## 1. Introduction

Vulvar cancer is a rare cancer whose incidence has increased in recent years. [1]. The most common type of vulvar cancer is squamous cell carcinoma (SCC), making up 70% of cases. The risk factors vary depending on whether the cancer is associated with human papillomavirus virus (HPV) infection. Age may increase the risk of vulvar cancer in non-HPV vulvar cancers. The International Federation of Gynaecology and Obstetrics (FIGO) has incorporated imaging findings into vulvar cancer staging [2]. MRI is the preferred imaging method due to its excellent soft-tissue resolution. It allows for correcting the local staging of vulvar cancer by assessing the involvement of adjacent tissue. The updated FIGO classification for vulvar cancer can assist clinicians in properly managing patients.

The primary treatment is surgical resection of the primary tumour with selective sentinel node biopsy (SNB) and/or bilateral inguinofemoral lymph node dissection (ILND) [3,4,5]. Postoperative recurrence rates are high, even in early-stage disease when tumour-free surgical margins are achieved [6,7] or in the absence of associated risk factors (lymph node metastases, deep stromal invasion, or lymphatic vascular space invasion [8,9,10,11]. Adjuvant treatment for patients with early-stage disease is a matter of debate due to the limited evidence available. The definition of tumour-free margins is not agreed upon, and there is uncertainty about the primary factors affecting the recurrence risk.

Treatment with radiotherapy is useful in patients with vulvar cancer both after surgery if risk factors are present and in the primary treatment of locally advanced tumours either exclusively or in combination with chemotherapy [12,13,14,15]. RT is a therapeutic option for the treatment of both the primary tumour and recurrences. In a recurrence setting, where surgery is not an option, brachytherapy (BT) alone or in combination with EBRT can be used to retreat lesions while preserving the patient’s functional status. 

Compared to EBRT, BT has the radiobiological potential to improve the dose to the target volume and minimize the dose to organs at risk. It is a more reproducible technique and covers the skin better than EBRT, without the need for a bolus. The rapid dose fall-off allows for the sparing of healthy organs such as the urethra and anus, which can be difficult to spare. A higher RT total dose on the lesion site may lead to better local control and a lower risk of disease progression [11,16]. Hypofractionated-accelerated treatment can also be performed using BT.

Most studies describing brachytherapy treatment of vulvar cancer are retrospective studies [17,18,19,20,21]. They include both adjuvant treatment and primary treatment of LAVC and recurrences. This review presents recent data on the role of BT in the treatment of primary and/or recurrent vulvar cancer including radiobiological, clinical, and therapeutic aspects.

## 2. Methods

This is a narrative review. The pubmed database analysed included publications published between 1990 and 2023. The terms used in the search included “vulvar cancer, brachytherapy, radiotherapy, skin toxicity, urethral toxicity, radiobiology”. We have restricted the search to articles published in English. Because vulvar cancer is a rare tumour, we included all types of publications in the search, including case series. The final analysis included 11 publications of patients with vulvar cancer treated with brachytherapy, as an exclusive treatment, after external beam radiation therapy or in recurrences.

## 3. Brachytherapy for Vulvar Cancer

### 3.1. Clinical Recommendations

Vulvar cancer is a rare tumour. There are few randomized studies and treatment recommendations are based mostly on retrospective studies. Different groups have published their consensus guidelines on the treatment of vulvar cancer [22,23,24,25].

It is widely acknowledged that age increases the risk of vulvar cancer. Treating cancer in older patients is challenging due to health status heterogeneity and treatment toxicity [26]. A very important aspect in patients with vulvar cancer is to individualize treatment, taking into account patient age and fitness levels. A geriatric assessment is crucial in determining the best treatment approach within a multidisciplinary setting. The Comprehensive Geriatric Assessment (CGA), first described in 1989 [27], is considered one of the cornerstones of geriatric care. This multidimensional tool measures frailty by assessing multiple domains, including functional and psychological status, comorbidities, cognitive function, social support, polypharmacy, and nutrition [28]. Recent data have shown that CGA is associated with side effects, morbidity, and mortality during cancer treatment [29]. This supports the prognostic and predictive value of oncological geriatric assessment. It is recommended by scientific societies to conduct a pre-treatment CGA in oncology practice to evaluate the potential risks and benefits. There are specialized working groups that focus on geriatric oncology research and aim to improve cancer decision-making for older patients. These groups analyse the unique vulnerabilities of elderly patients and work to guide personalized treatment plans that prioritize both outcomes and quality of life.

Surgery is the main treatment for early-stage vulvar cancer which is becoming a standard and includes the surgical resection of the primary tumour with selective sentinel node biopsy (SLNB) and/or bilateral inguinofemoral lymph node dissection (LND) [3,4,5]. Despite this, it is necessary to know the most important risk factors for locoregional recurrence. This would help to better determine if adjuvant treatment is necessary to reduce possible relapses. The two variables most strongly associated with recurrence are the status of the margins—positive or even close—and lymph node involvement [6,7,9]. Other variables can also affect relapses and determine adjuvant radiotherapy, such as stromal invasion or lymphovascular invasion [6,7,9]. Zapardiel et al. [7] showed an increased risk of recurrence in relation to lymph node involvement and resection margin in the VULCAN study—an international, multicentre, and retrospective study. This retrospective study showed that one of the factors significantly associated with an increase in local recurrence of squamous cell vulvar cancer was not undergoing radiotherapy. A recent review [30] showed that patients with lichen sclerosis, differentiated vulvar intraepithelial neoplasia (dVIN), and high-grade squamous intraepithelial lesions (HSIL) had a higher risk of local recurrence at 10 years.

Surgery is not feasible in all clinical situations. In patients with LAVC, either because it would require mutilating surgery and/or because of the presence of involved nodes attached to fascia, muscles, or vascular structures, the recommended treatment is radiotherapy, either combined or not combined with chemotherapy [14,15].

Brachytherapy could be used in the following scenarios: (1) as an adjuvant treatment after surgery for early-stage disease with unfavourable histological factors, either alone or after EBRT, (2) as a boost to the primary tumour after EBRT in patients with LAVC who are not candidates for surgery, and (3) in cases of relapse after surgery or previous radiotherapy.

Brachytherapy is a treatment modality with great advantages from the radiobiological point of view. It allows high doses to be delivered to the tumour and low doses to adjacent organs at risk. The integration of new imaging methods in the planning of brachytherapy further enhances this capability. Classically, most of the experience is based on low dose rate studies (LDR). LDR has now been replaced by pulsed dose rate (PDR) or high dose rate (HDR), and evidence suggests equivalence in terms of tumour control. In vulvar cancer there is no experience comparing LDR and PDR or HDR. PDR would theoretically have the physical advantages of HDR while retaining the radiobiological advantages of LDR. In general, PDR is administered in hourly fractions of a few minutes to achieve doses similar to those of LDR. According to the linear-quadratic model, the toxicity of late-response tissues (low a/b ratio) would be higher at higher doses per fraction or higher dose rates. From this point of view, LDR could be radiobiologically superior to HDR. However, HDR has the advantage of a better dose distribution that allows limiting doses to healthy tissues. A recent randomised study comparing HDR versus LDR has shown equal tumour control and less rectal toxicity for HDR [31]. Although PDR may be equivalent to LDR for acute and late effects [32], better logistics and dosimetric planning of HDR have caused PDR to be used less.

There are only a few retrospective studies that have examined the use of brachytherapy in treating vulvar cancer, including patients with early-stage, locally advanced, and recurrent disease. Due to the limited available evidence, the use of brachytherapy in patients with early-stage disease or locally advanced disease is controversial, making it difficult to obtain reliable data.

In a recent systematic review, Lancellota et al. [9] described 129 patients with vulvar cancer treated with brachytherapy alone or combined with EBRT. Data showed a median 5-year local control rate of 43.5% (range 19–68%) and a median 5-year overall survival rate of 50% (range 27–85%). Pohar et al. examined 34 patients treated with LDR-BT for a vulvar cancer treated either initially (*n* = 21) or in cases of recurrent disease (*n* = 13). With this heterogeneity of cases, they report acceptable local control rates [17]. Mahantshetty et al. included 38 patients treated with brachytherapy. And despite including patients with early operated tumours, LAVC tumours, or recurrences, they report high rates of local control [18]. A study conducted by Castelnau-Marchand et al. involved 26 patients who underwent BT [19]. The estimated DFS at 3 years was 57% (95% confidence interval [CI]: 45–69%). Eleven patients (42%) experienced tumour relapse. Ten patients experienced a local relapse as the first event (38% of the total. From the National Cancer Institute’s Surveillance, Epidemiology, and End Results (SEER), Rao et al. analysed 617 patients treated with EBRT alone and 32 patients received EBRT combined with BT [20]. The authors demonstrated an impact on disease-free survival in patients with lymph node involvement and advanced stages.

Approximately 40% of patients will experience recurrence, being more frequent during the two years after treatment (40–80%). Recurrent vulvar carcinoma has a poor prognosis, with 5-year survival rates ranging from 50% to 66% [1]. The location of the recurrence has important prognostic implications, with isolated local recurrences associated with a 5.6-fold increased risk of death; in cases with both local and nodal recurrences, the mortality risk is 14 times higher (HR, 14.1) [33]. Clinical examination is essential in routine surveillance, but imaging plays a crucial role [2,34]. Early detection of recurrence aids in better stratification of patients and the prompt initiation of therapies to improve outcomes and survival. PET/CT is superior to conventional imaging in this regard, as it assists in evaluating the treatment response and detecting disease recurrence [35]. Treatment for recurrence depends on the location of the relapse and on the treatment previously delivered. In cases of recurrences after exclusive surgery, the treatment will be the same as for de novo tumours. In recurrences after previous irradiation, the treatment must be individualized due to its complexity. In patients that are not candidates for surgery, reirradiation is a therapeutic option. If reirradiation is considered, EBRT or HDR-brachytherapy are possible options [12,17,18,19,21]. Brachytherapy is considered the preferred treatment in patients with comorbidities or not suitable for surgery. As explained above, one of the advantages of brachytherapy is that it allows for high doses of radiation to be administered to tumour tissue with minimal damage to the surrounding organs at risk. Lancellota et al. [9] showed the results of three studies accounting for 48 patients with vulvar recurrences treated with brachytherapy and/or EBRT. Studies have shown that 5-year local control rates range from 33–80%, although the data are not conclusive. In a recent study, Yaney et al. included eighteen patients, of whom twelve (66.7%) had primary disease of the distal vagina and vulva. The survival data reported by the authors does not discriminate between primary tumours of the vulva and recurrences of the lower third of the vagina [16]. Kellas-Ślęczka et al. evaluated 8 recurrent vulvar cancers after previous radical surgery. The estimated 1-year disease-free survival rate was 33% and 80% in LAVC and recurrences after surgery, respectively [21]. Pohar et al. examined 13 patients treated with LDR–BT for recurrent disease. The estimated local control rate at 5 years, was 47% (95% CI: 23–73%) [17]. Laliscia et al. identified 56 patients with recurrent vulvar cancer after primary surgery treated with RT (43 patients with EBRT and 13 patients with BT alone) [12]. Patients with local recurrence had a better prognosis than those with other sites of relapse. In the univariate analysis, there were no differences in DFS or OS (p NS) between EBRT and BT for these patients. Table 1 summarizes the larger series published in the literature.

### 3.2. Brachytherapy Technique

Vulvar tumours are commonly originated in the skin of the labia vulvae; around 70% of tumours involve the labia majora and labia minora, while only 15–20% affect the clitoris. In addition, radiation tolerance levels for some adjacent at-risk organs such as the clitoris, distal urethra, or skin are unknown. As a result, brachytherapy implants must be tailored to each patient and their specific clinical situation based on the location and extent of the disease. Prior to each implant, it is crucial to perform a thorough physical examination before each implant to identify and contour the full extent of the disease before brachytherapy implantation. However, there is currently no consensus on which imaging techniques are best. Pre-treatment MRI is particularly important in revealing the extent of the tumour and the degree of the involvement of adjacent organs [2].

The brachytherapy technique for vulvar cancer consists of an interstitial implant with either rigid needles or plastic tubes. The latter option is more comfortable for the patient. In tumours with extensive vaginal involvement, a multichannel vaginal cylinder or another intracavitary vaginal applicator can be helpful. The use of transperineal implants combined with the vulvar interstitial implant is also recommended. Fiducial markers are placed to allow for visualisation and contouring of the target volume on CT scans (GTV or surgical bed).

Additionally, after the excision of a recurrence, intraoperative brachytherapy [36,37] or intraoperative radiotherapy [38] can be administered, although the evidence is scarce (mainly case reports).

Unlike other gynaecological tumours (cervical cancer, endometrial cancer, and vaginal cancer) [39,40,41,42], there is currently no international consensus on the nomenclature of BT target volumes in vulvar cancer. Yaney et al. suggest using the high-risk clinical target volume (CTV-Thr) at the time of brachytherapy implantation, following the terminology used in gynaecological interstitial brachytherapy [16,43]. If there is visible macroscopic disease present at the time of BT (GTVres), it may be contoured based on physical examination (and marked with fiducial markers) and/or imaging findings (MRI), although there is no consensus on this aspect. Following these recommendations, the GTV will be delimited at the time of diagnosis if BT is the first treatment, or the residual volume (GTVres) in case brachytherapy is performed after RT. The CTV-Thr, will include the GTV with an additional margin MRI can be useful in the contouring [43].

The applicator is reconstructed based on CT or MRI images. Following ICRU 58 recommendations [44], doses are prescribed at the reference isodose (85% of the minimum dose rate between planes) according to Paris system criteria and expressed as biological equivalent dose (BED) of 2 Gy/fraction (BED2Gy) (a/b = 10 Gy to the tumour, half-time of repair) (Figure 1). Different fractionated BT regimens are currently in use, and the available literature is based on retrospective experiences [17,18,19,20,21].

A higher total dose of RT at the lesion site may be associated with better local control and a lower risk of disease progression [18,19,20]. Brachytherapy has an advantage over EBRT in delivering a higher dose over the target volume while minimizing the dose to OARs. There is no evidence in the literature that dose escalation with brachytherapy improves local control of vulvar cancer. However, when analysing external radiotherapy series, greater local and lymph node control has been related to higher doses on the primary tumour [11].

In recurrences with previous radiotherapy, BT treatment should be individualized. The brachytherapy dose should be expressed as BED of 2 Gy/fraction (BED2Gy) always considering the BED2Gy from the previous treatment and the time elapsed between the two treatments. However, published data on this area is very limited.

The different OARs to be considered in brachytherapy for vulvar cancer are the skin, the distal vaginal mucosa, the urethra, the anus, and the clitoris. There is limited published data on the potential toxicity of brachytherapy treatment for vulvar cancer and the limitations of organs at risk (OARs) in this clinical entity. Skin tolerance is related to the dose administered, the implant volume, and the V150 and V200 volumes. The main late complication is ulceration or necrosis although there is no clear consensus in the literature regarding dose limits. Dyk et al. relate G3-4 toxicity to high V100 volumes [45]. The late vaginal toxicities comprise vaginal dryness, dyspareunia, and vaginal stenosis. However, the dose-volume parameter to determine vaginal toxicity is unclear, especially in relation to the lower third of the vagina. The urethra is one of the most important OARs in this anatomical location. A Canadian study [46] evaluated 83 patients undergoing brachytherapy for vaginal cancer. That study demonstrated that different dose levels are predictors of urethral toxicity.

In this setting, one important field for the application of artificial intelligence (AI) in healthcare is RT. Theoretically, AI methods could improve the quality of radiotherapy treatment, by assisting in optimizing the applicators’ location in BT planning and the optimal source position in targets, while avoiding the irradiation of OARs. In this sense, the adoption of AI methods for dosimetric parameters related to the different OARs and their correlation with clinical parameters could be very helpful.

## 4. Conclusions and Future Directions Section

Vulvar cancer treatment is highly challenging due to the lack of robust evidence supporting available treatment options. This review provides specific recommendations to help clinicians with the technical aspects of using brachytherapy to treat vulvar cancer, both treatment of the primary tumour and recurrences.

Brachytherapy is a promising treatment for vulvar cancer due to its radiobiological properties. Despite its declining use, the application of artificial intelligence in the treatment process may lead to a resurgence of brachytherapy as an effective treatment option for this type of tumour in the coming years. The results of this review emphasize the importance of conducting clinical trials to establish the best approach for managing this uncommon disease.

## Figures and Tables

**Figure 1 cancers-15-05581-f001:**
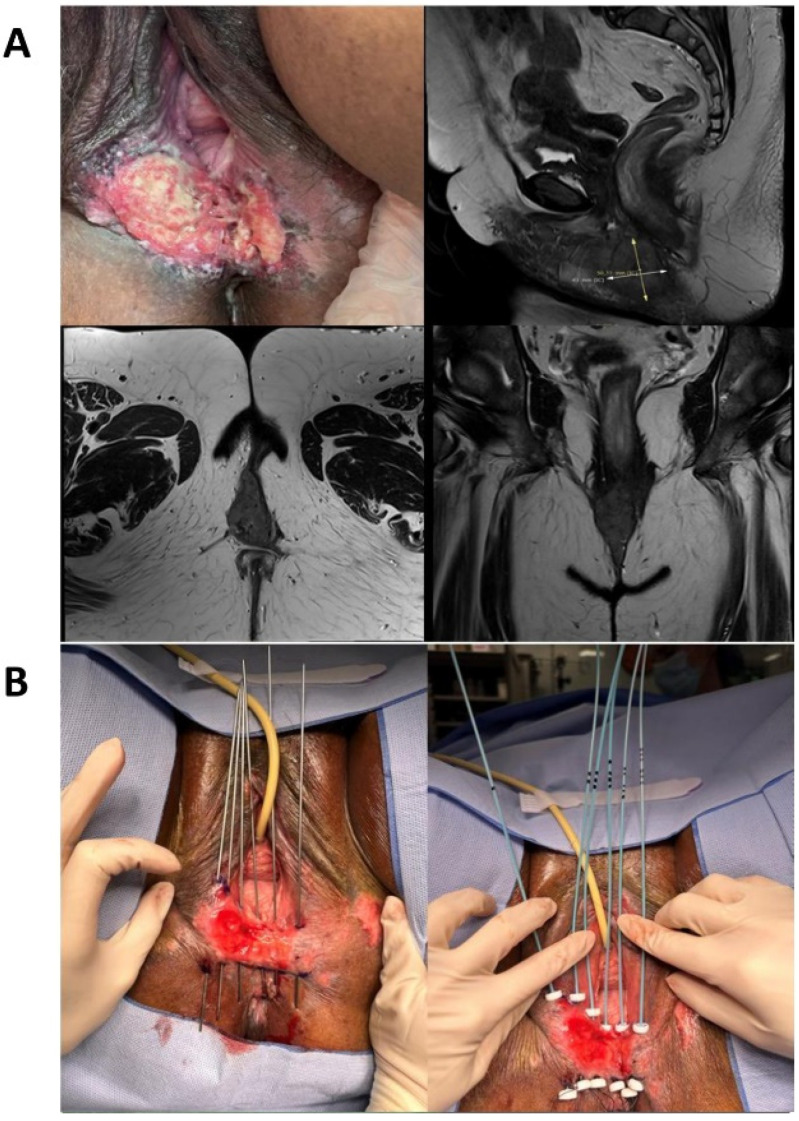
Example of a vulvar brachytherapy implant. A case of a 60-year-old patient with an exophytic lesion with irregular edges, partially necrotic, hard and friable to the touch, measuring approximately 7 × 6 cm^2^, affecting the left labium majus to the introitus with perineal extension. (**A**) Clinical examination and diagnostic imaging (MRI). (**B**) Interstitial implant with rigid needles and definitive plastic tubes. (**C**) CT reconstruction of catheters, dosimetric distribution in axial and sagittal planes, and dose-volume histogram.

**Table 1 cancers-15-05581-t001:** Literature review.

Author	Year/Follow-Up	Patients	Outcomes	Comments
Yaney et al. (2021) [16]	2012–2019. Mean follow-up: 15.6 m	LAVC: 10 REC: 1	2-year DFS: 75% 2-year OS: 73.4%	This study reports high local control in tumours of the vulva and distal third of the vagina treated with brachytherapy.
Laliscia et al. (2019) [12]	1992–2016	REC: 56	5-year DFS: 19% 5-year OS: 43%	
Pohar et al. (1995) [17]	1975–1993	LAVC: 21 REC: 13p	LAVC: 5-year LC:80% 5-year OS:27% REC 5-year LC: 19% 5-year OS:33%	The authors reflect in this series that in cases that are not candidates for surgery, brachytherapy alone may offer good results, especially in small tumours.
Mahantshetty et al. (2017) [18]	2001–2016. Mean follow-up: 30 months	LAVC: 29 ADJ: 6 REC: 3	RT + BT: 5-year LC:68% 5-year DFS: 44% 5-year OS: 85% BT: 5-yearLC:100%, (NS) 5-year DFS: 80% 5-year OS: 80%	The authors conclude that treatment with BT alone had better, non-significant results, probably because smaller lesions were treated with brachytherapy alone and larger lesions with combined treatment. This could be explained by the fact that early lesions have a better prognosis compared to more advanced vulvar lesions.
Castelnau-Marchand et al. (2017) [19]	2000–2015. Mean follow-up: 41 months	LAVC: 8 ADJ: 15 REC: 3	3-year DFS: 37% 3-year OS: 81%	The authors note the low toxicity rates in their series, both acute and delayed, probably related to the rigorous selection of patients and the centralization of such rare and complex treatments in comprehensive cancer centres.
Rao et al. (2017) [20]	1973–2011	LAVC: 649	RT + BT: 5-year DFS: 45% 5-year OS: 34% BT: 5-year DFS: 33% 5-year OS: 24%	In this review by the American SEER group, the authors demonstrated that combined treatment with BT is not associated with improved survival compared to EBRT alone although certain subgroups of patients may benefit from brachytherapy, but this hypothesis requires validation in future studies.
Kellas-Ślęczka, 2016 [21]	2004–2014 Mean follow-up: 12 months	LAVC: 6 REC: 8	RT + BT: 1-year DFS: 33% 1-year OS: 80% BT: 1-year DFS: 80% 1-year OS: 100%	The authors report that in their series, patients with advanced primary disease were older with severe comorbidities and the vast majority of treatments were palliative in intent. However, patients with recurrent disease were younger and were on regular follow-ups after previous treatment. They observed significant differences in OS according to the median V100.

LAVC: locally advanced vulvar cancer, ADJ: postoperative treatment, REC: recurrences. DFS: disease free survival, LC: local control, OS: overall survival.

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
