# Peer review of "Is There a Place for Brachytherapy in Vulvar Cancer? A Narrative Review"

_cancers, 2023, doi:10.3390/cancers15235581_

Round 1

Reviewer 1 Report

Comments and Suggestions for Authors

A narrative or traditional literature review is a comprehensive, critical, and objective analysis of the current knowledge on a given topic. While it is an essential part of the research process and helps to establish a theoretical framework and focus or context of research, there is no consensus on the standard structure of a narrative review. The preferred approach is the IMRAD (Introduction, Methods, Results, and Discussion) structure.

One of the primary concerns with the present narrative review paper is the lack of defining the IMRAD  design.

The authors have not clearly outlined the methodology of their literature research. As a result, the manuscript appears to be a collection of closely connected information of literature findings.

The flow of ideas is unclear, and there is a lack of coherence between sections.

The paper should have presented with a clear introduction, outlining the main themes or categories of treatments and providing a concise summary or conclusion to the content.

Additionally, sections like "Radiobiology of Brachytherapy" and "Brachytherapy Techniques"  fit more into an educational article as a narrative review about brachytherapy of vulvar cancer.

Furthermore, the title "Is there a place for brachytherapy in Vulvar Cancer? State of the art." suggest a comparison with other treatment methods - which is involved in the present manuscript.

Author Response

1. Summary

Although it may not be the most appropriate, we attach the new manuscript with the suggested changes for a better understanding. All pages and lines referred to in the replies to comments refer to this new version. Please let me know if this methodology is not appropriate.

Comments 1: A narrative or traditional literature review is a comprehensive, critical, and objective analysis of the current knowledge on a given topic. While it is an essential part of the research process and helps to establish a theoretical framework and focus or context of research, there is no consensus on the standard structure of a narrative review. The preferred approach is the IMRAD (Introduction, Methods, Results, and Discussion) structure.

One of the primary concerns with the present narrative review paper is the lack of defining the IMRAD design.

Response 1:

As advised by the reviewer, we have structured the manuscript with an introduction, methods and discussion of the evidence in the literature regarding brachytherapy treatment, and final conclusions.

We have introduce a “Methods”.

Page:2, Line 69.

“ [Methods

This is a narrative review. The pubmed database analyzed included between 1995 and 2023. The terms used in the search included "vulvar cancer, brachytherapy, radiotherapy, skin toxicity, urethral toxicity, radiobiology". We have restricted the search to articles published in English. Because vulvar cancer is a rare tumour we have included in the search all types of publications including case series. The final analysis included 11 publications of patients with vulvar cancer treated with brachytherapy, as exclusive treatment, after external beam radiation therapy or in recurrences.]”

As advised by the reviewer in comment 5, we have reduced the block on radiobiology; and instead of writing general concepts, we have integrated it into the text under "clinical considerations of brachytherapy of vulvar cancer".

Page 3: line 124.

“[Brachytherapy is a treatment modality with great advantages from the radiobio-logical point of view. It allows high doses to be delivered to the tumour and low doses to adjacent organs at risk. The integration of new imaging methods in the planning of brachytherapy further enhances this capability. Classically, most of the experience is based on low dose rate studies (LDR). LDR has now been replaced by pulsed dose rate (PDR) or high dose rate (HDR), and evidence suggests equivalence in terms of tumour control.

In vulvar cancer there is no experience comparing LDR and HDR……….]”

We have integrate the block on Comprehensive Geriatric Assessment into the text under "clinical considerations of brachytherapy of vulvar cancer".

Page 2: line 82.

[“It is widely acknowledged that age increases the risk of vulvar cancer. Treating cancer in older patients is challenging due to health status heterogeneity and treat-ment toxicity (26). A very important aspect in patients with vulvar cancer is to indi-vidualize treatment, taking into account patient age and fitness levels……”]

Comments 2: The authors have not clearly outlined the methodology of their literature research. As a result, the manuscript appears to be a collection of closely connected information of literature findings.

Response 2:

As advised by the reviewer, we have elaborated a discussion of the evidence in the literature regarding brachytherapy treatment which includes both “clinical recomendations” and “brachytherapy technique” instead of independent information.

Page 2: line 77 and line 78.

[“Brachytherapy for vulvar cancer.

1.         Clinical Recommendations”]

Page 5: line 199.

[“2.      Brachytherapy Technique”]

Comments 3: The flow of ideas is unclear, and there is a lack of coherence between sections.

Response 3:

With the changes mentioned in Question 1 and question 2, we believe that the manuscript have an IMRAD design and the sections are integrated with coherence between the different them.

Comments 4: The paper should have presented with a clear introduction, outlining the main themes or categories of treatments and providing a concise summary or conclusion to the content.

Response 4:

See previous comments.

Comments 5: Additionally, sections like "Radiobiology of Brachytherapy" and "Brachytherapy Techniques" fit more into an educational article as a narrative review about brachytherapy of vulvar cancer.

Response 5: Agree. We have modified and reduced the chapter about “radiobiology” and we have integrated “the brachytherapy technique”.

See previous comments.

Comments 6: Furthermore, the title "Is there a place for brachytherapy in Vulvar Cancer? State of the art." suggest a comparison with other treatment methods - which is involved in the present manuscript.

Response 6: Agree.

As advised by the reviewer, we have modified the title. The new one is: "Is there a place for brachytherapy in Vulvar Cancer? A narrative review."

Reviewer 2 Report

Comments and Suggestions for Authors

Interestingly and import review to show that BT could be a good alternative for EBRT or surgery, however I miss some essential information, and also the order of the paragraphs are to my opinion not always logical. Finally, the paper would be more interesting for the reader if an example of a BT treatment could be included in a Figure.

Find below my suggestions for further approvement of the paper (in order of appearance):

- is the abbreviation locally advanced vulvar cancer not LAVC instead of LACV (abstract and throughout manuscript)?

- I would like to suggest to drastically concise the chapter about radiobiology of BT. To my opinion, all these formulas doesn't add much for the reader who would like to do BT for vulvar cancer (maybe as supplement?). What is missing in this paragraph is the explanation about the differences between LDR, PDR and HDR (maybe in a Figure?)

- please explain the concept of HR CTV for the audience of this journal. To my opinion this is more importantly then all the formulas for BT

- can you give a recommendation when pts are referred to a specialist in elderly patients when the CGA shows a high score on frailty? Not all pts are old and fragile, and what is the definition of old?

- I disagree with the authors that BT is indicated in postoperative setting with no macroscopic tumor. There are data that EBRT is the first choice and according to my knowledge there is no literature about applying BT in post op setting. I however agree that BT can be applied as a boost in primary setting or in case of a recurrence in a former irradiated location.

- a short paragraph about radiobiology fits probably better in the paragraph about clinical recommendations for vulvar BT

- in case of reirradiation not only HDR, but also PDR treatment is suitable, and I believe that PDR might even better (in RB perspective) than HDR

- sentence 266 "In tumors with extensive vaginal involvement, the use of trans perineal implants combined with the vulvar interstitial implant is recommended". I would also add that in these cases a multichannel vaginal cilinder  or another intracavitary vaginal applicator can be helpful.

- regarding the nomenclature of the targets, I would like to suggest to use the nowadays used GEC ESTRO nomenclature, namely CTV-Thr, CTV-Tir en GTV-Tres

- MRI can also be used for applicator and needle reconstruction, beside CT

- sentence 266: 'mean time' 1.5 h is not the correct nomenclature; it is half-time of repair. Something which is important to explain in the RB section when discussing PDR.

- sentence 302-304. The reference to published guidelines are incorrect/not existing. As also stated earlier by the authors, no guidelines for BT +/- do exist. Table 2 is there for misleading. Although it is good to have some guidance, the only thing what can be stated here is which dose scheduled are used. In addition, I miss the total dose to aimed for in EQD2.

-although data is limiting, I would like to suggest to add a little bit more about toxicity after irradiation of the vulva, especially when treated with BT. I think that besides urethral toxicity, ulceration of the skin and distal vaginal mucosal tumors is as important as urethral toxicity.

- it would be illustrative for the reader to show an example of vulvar BT with images (preferably with MRI/photos of the tumor at diagnosis and at BT including visualization of the implanted catheters on MRI and/or CT

- regarding the literature references: please double check if all papers are correctly referred to, and at the correct place. I also miss a short paragraph about how literature search has been performed.

Comments on the Quality of English Language

no specific comments.

Author Response

1. Summary

Although it may not be the most appropriate, we attach the new manuscript with the suggested changes for a better understanding. All pages and lines referred to in the replies to comments refer to this new version. Please let me know if this methodology is not appropriate.

Comments 1: Is the abbreviation locally advanced vulvar cancer not LAVC instead of LACV (abstract and throughout manuscript)?

Response 1: Agree. I have change LAVC instead of LACV.

1.       Abstract: line 25 “Adjuvant treatment after surgery as well as primary treatment of locally advanced vulvar cancer (LAVC) are two key radiotherapy treatment scenarios, external beam radiation therapy (EBRT) combined or not with brachytherapy (BT).”

2.       Page 2: line 66. “For patients with LAVC, external beam radiotherapy (EBRT) or chemoradiation, either neoadjuvant or radical, are the preferred treatments.”

3.       Page 3: line 115. “Surgery is not feasible in all clinical situations. In patients with LAVC, either because”

4.       Page 3: line 121. “as a boost to the primary tumour after EBRT in patients with LAVC who”

5.       Page 3: line 153. “operated tumours, LAVC tumours or recurrences, they report high rates of local control”

Comments 2: I would like to suggest to drastically concise the chapter about radiobiology of BT. To my opinion, all these formulas doesn't add much for the reader who would like to do BT for vulvar cancer (maybe as supplement?). What is missing in this paragraph is the explanation about the differences between LDR, PDR and HDR (maybe in a Figure?)

Response 2: Agree. We have modified the chapter about radiobiology.

As advised by the reviewer, we have reduced the block on radiobiology; and instead of writing general concepts, we have integrated it into the text under "clinical considerations of brachytherapy of vulvar cancer".

Page 3: line 124. See comment 6.

Comments 3: Please explain the concept of HR CTV for the audience of this journal. To my opinion this is more importantly then all the formulas for BT

Response 3: Agree.

As we have reduced the block of radiobiology, the term of HRCTV is now introduced later.

Page 6. Lines 222.:

“[Yaney et al. suggest using the high-risk clinical target volume (CTV-Thr) at the time of brachytherapy implantation, following the terminology used in gynaecological interstitial brachytherapy [16,43].]”

See comment 9.

Comments 4: Can you give a recommendation when pts are referred to a specialist in elderly patients when the CGA shows a high score on frailty? Not all pts are old and fragile, and what is the definition of old?

Response 4: Agree.

Not all patients are old and fragile. However, in elderly and frail vulvar cancer patients, we should probably adapt the treatment to their overall condition in order not to worsen their quality of life.

Nevertheless, education on frailty could greatly assist radiation oncologists to adapt treatment according to older patient’s needs (for example, one of the aspects that geriatric assessment might alter is the omission of concomitant chemotherapy). Actually, the CGA has been considered the gold standard for the identification of frailty. We can use another frailty screening tools (p.E G8) but no consensus exists on the best screening tool to use in radiation oncology.

There is no consensus on “what is the definition of old”. The Age Cut-Off in several studies of CGA in Radiation Oncology is >65, >70 and >75years.

Comments 5: I disagree with the authors that BT is indicated in postoperative setting with no macroscopic tumor. There are data that EBRT is the first choice and according to my knowledge there is no literature about applying BT in post op setting. I however agree that BT can be applied as a boost in primary setting or in case of a recurrence in a former irradiated location.

Response 5: We have, accordingly, revised to emphasize this point.

Probably the main indications for brachytherapy are LAVC after external radiotherapy and recurrences. However, there are special situations where exclusive brachytherapy could be a very useful technique in adjuvant treatment with positive margins as reflected by other authors and seen in table 1 (MAHANTSHETTY et al., (2017) and CASTELNAU-MARCHAND et al., (2017)).]

We have change the sentence as follows:

Page 3: line 119.

“[Brachytherapy could be used in the following scenarios: 1) as an adjuvant treatment after surgery for early-stage disease with unfavourable histological factors, either alone or after EBRT, 2) as a boost to the primary tumour after EBRT in patients with LAVC who are not candidates for surgery and 3) in cases of relapse after surgery or previous radiotherapy.]”

Comments 6: a short paragraph about radiobiology fits probably better in the paragraph about clinical recommendations for vulvar BT

Response 6: Agree. We have modified the “radiobiology” to emphasize this point.

As advised by the reviewer, we have reduced the block on radiobiology; and instead of writing general concepts, we have integrated it into the text under "clinical considerations of brachytherapy of vulvar cancer".

Page 3: line 124.

“[Brachytherapy is a treatment modality with great advantages from the radiobio-logical point of view. It allows high doses to be delivered to the tumour and low doses to adjacent organs at risk. The integration of new imaging methods in the planning of brachytherapy further enhances this capability. Classically, most of the experience is based on low dose rate studies (LDR). LDR has now been replaced by pulsed dose rate (PDR) or high dose rate (HDR), and evidence suggests equivalence in terms of tumour control.

In vulvar cancer there is no experience comparing LDR and HDR……….]”

Comments 7:  in case of reirradiation not only HDR, but also PDR treatment is suitable, and I believe that PDR might even better (in RB perspective) than HDR

Response 7: Agree. We have revised and added some evidences to emphasize this point.

Page 3: line 130.

“[PDR would theoretically have the physical advantages of HDR while retaining the radio-biological advantages of LDR. In general, PDR is administered in hourly fractions of a few minutes to achieve doses similar to those of LDR.  According to the linear-quadratic model, the toxicity of late-response tissues (low a/b ratio) would be higher at higher doses per fraction or higher dose rates.  From this point of view, LDR could be radiobiologically su-perior to HDR. However, HDR has the advantage of a better dose distribution that allows limiting doses to healthy tissues. A recent randomised study comparing HDR versus LDR has shown equal tumour control and less rectal toxicity for HDR [31]. Although PDR may be equivalent to LDR for acute and late effects [32] better logistics and dosimetric planning of HDR have made PDR less used.]”

Comments 8: sentence 266 "In tumors with extensive vaginal involvement, the use of trans perineal implants combined with the vulvar interstitial implant is recommended". I would also add that in these cases a multichannel vaginal cilinder  or another intracavitary vaginal applicator can be helpful.

Response 8: Agree. We have modified the text to emphasize this point.

Page 5.: line 210.

“[The brachytherapy technique for vulvar cancer consists of an interstitial implant with either rigid needles or plastic tubes. The latter option is more comfortable for the patient. In tumours with extensive vaginal involvement, a multichannel vaginal cylinder or another intracavitary vaginal applicator can be helpful. The use of transperineal implants combined with the vulvar interstitial implant is also recommended. Fiducial markers are placed to allow for visualisation and contouring of the target volume on CT scans (GTV or surgical bed).]”

Comments 9: regarding the nomenclature of the targets, I would like to suggest to use the nowadays used GEC ESTRO nomenclature, namely CTV-Thr, CTV-Tir en GTV-Tres.

Response 9: Agree. We have modified the text and added this point.

Page 6. Lines 220:

“[Unlike other gynaecological tumours (cervical cancer, endometrial cancer and vagi-nal cancer) [39-42], there is currently no international consensus on the nomenclature of BT target volumes in vulvar cancer. Yaney et al. suggest using the high-risk clinical target volume (CTV-Thr) at the time of brachytherapy implantation, following the terminology used in gynaecological interstitial brachytherapy [16,43]. If there is visible macroscopic disease present at the time of BT (GTVres), it may be contoured based on physical exami-nation (and marked with fiducial markers) and/or imaging findings (MRI), although there is no consensus on this aspect.. Following these recommendations, the GTV will be delimited at the time of diagnosis if BT is the first treatment, or the residual volume (GTVres) in case brachytherapy is performed after RT. The CTV-Thr, will include the GTV with an ad-ditional margin MRI can be useful in the contouring [43].]”

Comments 10: MRI can also be used for applicator and needle reconstruction, beside CT

Response 10: Agree. We have modified the text to emphasize this point.  

One important aspect in brachytherapy procedure is catheter reconstruction. Nevertheless, a clear identification of the source path inside the plastic tubes in T2W sequences could be difficult.

Page 6: line 231.

 “[The applicator is reconstructed, based on CT or MRI images.]”

Comments 11: sentence 266: 'mean time' 1.5 h is not the correct nomenclature; it is half-time of repair. Something which is important to explain in the RB section when discussing PDR.

Response 11: Agree. We have modified the text to change this point. 

Page 6: line 231.

“[Following ICRU 58 recommendations [44], doses are prescribed at the reference isodose (85% of the minimum dose rate between planes) according to the Paris system criteria and expressed as biological equivalent dose (BED) of 2 Gy/fraction (BED2Gy) (a/b=10 Gy to the tumour, half-time of repair) (Figure 1).]”

Comments 12: sentence 302-304. The reference to published guidelines are incorrect/not existing. As also stated earlier by the authors, no guidelines for BT +/- do exist. Table 2 is there for misleading. Although it is good to have some guidance, the only thing what can be stated here is which dose scheduled are used. In addition, I miss the total dose to aimed for in EQD2.

Response 12: Agree. We have modified the text to change this point. 

As the reviewer refers to the table 2, it could be misleading, especially if it is not specified in EQD2. In this case the table and the text referring to it should be deleted.

Page 6: line 235.

“[Different fractionated BT regimens are currently in used, and the available literature is based on retrospective experiences.]”

Table 2 shows the recommended doses for both exclusive BT and combined modality with EBRT treatment for primary tumour and recurrences, based on published guide-lines [34,36].

Comments 13: although data is limiting, I would like to suggest to add a little bit more about toxicity after irradiation of the vulva, especially when treated with BT. I think that besides urethral toxicity, ulceration of the skin and distal vaginal mucosal tumors is as important as urethral toxicity.

Response 13: Agree. We have modified the text to change this point. 

We have modified the text to include skin and vaginal toxicity data. And we have reduced the information regarding urethral toxicity.

Page 8: line 256.

“[The different OARs to be considered in brachytherapy for vulvar cancer are the skin, the distal vaginal mucosa, the urethra, the anus, and the clitoris. There is limited pub-lished data on the potential toxicity of brachytherapy treatment for vulvar cancer and the limitations of organs at risk (OARs) in this clinical entity. Skin tolerance is related to the dose administered, the implant volume, and the V150 and V200 volumes. The main late complication is ulceration or necrosis although there is no clear consensus in the literature regarding dose limits. Dyk et al. relate G3-4 toxicity to high V100 volumes [45]. The late vaginal toxicities comprise vaginal dryness, dyspareunia and vaginal stenosis. However, the dose-volume parameter to determine vaginal toxicity is unclear especially in relation to the lower third of vagina. The urethra is one of the most important OAR in this anatomical location.   A Canadian study [46] evaluated 83 patients undergoing brachytherapy for vaginal cancer. That study demonstrated that different dose levels are predictors of urethral toxicity.]”

Comments 14:  it would be illustrative for the reader to show an example of vulvar BT with images (preferably with MRI/photos of the tumor at diagnosis and at BT including visualization of the implanted catheters on MRI and/or CT.

Response 14: Agree. We have included a figure. 

We have included a figure (Figure 1. Example of a vulvar brachytherapy implant).

Page7: line 238.

“[Figure 1. Example of a vulvar brachytherapy implant.

A case of a 60-year-old patient with an exophytic lesion with irregular edges, partially necrotic, hard and friable to the touch, measuring approximately 7x6cm, affecting the left labium majus to the introitus with perineal extension. A. Clinical examination and diagnostic imaging (MRI). B. Interstitial implant with rigid needles and definitive plastic tubes. C. CT reconstruc-tion of catheters, dosimetric distribution in axial and sagittal planes and dose-volume histogram.]”

Comments 15:  regarding the literature references: please double check if all papers are correctly referred to, and at the correct place. I also miss a short paragraph about how literature search has been performed.

Response 15: Agree.

We have double checked all manuscript and correctly referred.

We have added a short paragraph about how literature search has been performed.

 Page 2: line 69

“[Methods:

This is a narrative review. The pubmed database analyzed included between 1990 and 2023. The terms used in the search included "vulvar cancer, brachytherapy, radiotherapy, skin toxicity, urethral toxicity, radiobiology". We have restricted the search to articles published in English. Because vulvar cancer is a rare tumour we have included in the search all types of publications including case series. The final analysis included 11 publications of patients with vulvar cancer treated with brachytherapy, as exclusive treatment, after external beam radiation therapy or in recurrences.]”

Round 2

Reviewer 1 Report

Comments and Suggestions for Authors

All reviewer queries were adequately addressed in the new version.